# Evaluation of Crocin Content and In Vitro Antioxidant and Anti-Glycation Activity of Different Saffron Extracts

**DOI:** 10.3390/plants12203606

**Published:** 2023-10-18

**Authors:** Simone Ronsisvalle, Annamaria Panico, Debora Santonocito, Edy Angela Siciliano, Federica Sipala, Lucia Montenegro, Carmelo Puglia

**Affiliations:** 1Department of Drug and Health Sciences, University of Catania, Viale Andrea Doria n°6, 95125 Catania, Italy; simone.ronsisvalle@unict.it (S.R.); panico@unict.it (A.P.); debora.santonocito@unict.it (D.S.); edysiciliano@hotmail.it (E.A.S.); sipalafederica@gmail.com (F.S.); capuglia@unict.it (C.P.); 2NANOMED-Research Center on Nanomedicine and Pharmaceutical Nanotechnology, University of Catania, 95125 Catania, Italy

**Keywords:** crocin, saffron extracts, antioxidants, carotenoids, anti-glycation activity

## Abstract

Crocin, a glycoside carotenoid that exhibits several health benefits, is mainly obtained from saffron (*Crocus sativus* L.), whose quality and content of phytochemicals can be strongly affected by environmental conditions. Therefore, in this work, the crocin content and in vitro antioxidant activity of saffron extracts obtained from three different varieties (Greek, Sicilian, and Iranian saffron) were assessed. Crocin content in saffron extracts was quantified via ultra-performance liquid chromatography coupled with mass spectrometry. The antioxidant activity of saffron extracts was evaluated using the oxygen radical absorbance capacity (ORAC) assay and nitric oxide (NO) radical scavenging test. The Maillard reaction was used to assess anti-glycation activity. Although the Sicilian and Iranian saffron extracts contained higher amounts of crocin (128 ± 6 ng/mL and 126 ± 4 ng/mL, respectively) compared to the Greek extracts (111 ± 2 ng/mL), ORAC values (50.9 ± 0.5) and % NO inhibition (35.2 ± 0.2) were higher for the Greek variety, which displayed a total phenolic content about two-fold greater than that of the other two extracts. Sicilian and Greek saffron had similar anti-glycation activities, while Iranian saffron was less effective. These results suggest that the antioxidant activity of saffron extracts could be ascribed to their naturally occurring complex mixture of phytochemicals, deserving further investigation as supplements to prevent pathological conditions induced by radical species.

## 1. Introduction

Crocin (all-*trans* crocetin di-β-D-gentiobiosyl ester) is a glycoside carotenoid (Figure 1) endowed with several health benefits, including antioxidant, anticancer, antidepressant, anxiolytic, cardioprotective, and neuroprotective activity [1]. Crocin belongs to the “crocin” family, consisting of hydrophilic carotenoids in which D-glucose and/or D-gentiobiose residues form either mono- or di-glycosyl polyene esters of crocetin. It is noteworthy that the high glycosyl content makes crocins unusual water-soluble carotenoids [2]. 

The main sources of these natural compounds are the dried stigmas of *Crocus sativus* L. (commonly named saffron) and the fruits of *Gardenia jasminoides* E. *C. sativus* L. is a perennial herb, belonging to the *Iridaceae* family, and is widely cultivated in the Mediterranean area and western Asia, with Iran being its main producer [3,4]. Due to its properties, saffron is one of the world’s most expensive and popular spices, widely used to add deep flavor and color to several kinds of dishes [5]. 

Today, saffron is being widely investigated for its potential therapeutic applications. Many studies highlighted the ability of saffron extracts to act as antioxidant, anti-inflammatory, anticancer, wound-healing, and anti-aging agents [6,7,8,9,10,11,12,13]. Clinical trials highlighted the efficiency of saffron capsules in ameliorating depression symptoms in comparison to placebo [14] or conventional drugs such as imipramine [15] and fluoxetine [16]. Similarly, clinical studies performed on patients with mild to moderate Alzheimer’s disease showed significantly better outcomes on cognitive function after administering saffron capsules in comparison to placebo [17], while the same effectiveness was observed when saffron capsules were compared to donezepril [18]. Studies carried out on different animal models demonstrated the efficiency of saffron and its active ingredients, crocin and safranal, in the treatment of pathological conditions such as atherosclerosis [19,20], cancer [21,22,23], hyperglycemia, and glucose uptake/metabolization [24]. It is interesting to note that Shirali et al. [25] reported the ability of crocin to significantly lower the formation of advanced glycation end-products, in addition to the levels of serum glucose, triglycerides, and total cholesterol in the diabetic rats. Goyal et al. [26] suggested the feasibility of using crocin as a cardioprotective agent due to the improvement of cardiac functions in Wistar albino rats after crocin pretreatment. At a cutaneous level, the involvement of crocin [27] and saffron extracts [28,29,30] in anti-inflammatory, antioxidant, and aging related processes has been reported. 

The quality of saffron and its content of phytochemicals, in which crocin plays a key role in determining antioxidant activity, can be strongly affected by cultivation, territorial and climatic conditions [31,32]. Hence, it is crucial to evaluate the concentration of active ingredients according to the different saffron varieties. 

In this context, the aim of this work was to assess the crocin content and the in vitro antioxidant activity of saffron extracts obtained from three varieties (Greek, Sicilian, and Iranian). The in vitro antioxidant activity of crocin has been already reported by others using different methods, such as DPPH (2,2′-diphenylpicryl hydrazyl free radical) radical scavenging assay [33], ethylene assay, and squalene peroxidation assay [30]. As DPPH is hydrophobic, the test requires the use of organic solvents [34], which have an adverse environmental impact. As crocin and saffron extracts are water-soluble, alternative methods, which do not require the use of organic solvents, could be carried out. The ethylene assay and the squalene peroxidation assay assess hydroxyl radical and singlet oxygen scavenging activity, respectively [30]. Both these methods were developed to demonstrate the antioxidant ability of topical formulations [35,36], and they were not validated on plant extracts. Therefore, we assessed the antioxidant activity of saffron extracts using the oxygen radical absorbance capacity (ORAC) assay, which is regarded as a reliable tool for screening antioxidants [34]. Additionally, in vitro nitric oxide (NO) radical scavenging ability of the saffron extracts under investigation was evaluated. As oxidative stress may result in the formation of advanced glycation end products (AGEs), a further aim of this study was to determine the anti-glycation activity of saffron extracts using the Maillard reaction. 

## 2. Results and Discussion

### 2.1. Crocin Determination in Saffron Extracts

According to the literature [1], different extraction methods can be used to obtain the components, including crocin, from saffron dried stigmas. Therefore, we carried out preliminary experiments to set out the optimal extraction conditions. The content of crocin in each saffron extract was determined by ultra-performance liquid chromatography coupled with mass spectrometry (UPLC-MS/MS) analysis and reported in Table 1. The Sicilian and the Iranian saffron contained similar amounts of crocin that were significantly higher (*p* < 0.05) than that of the Greek variety.

According to the literature [1], both geographical location and processing methods could affect the quality of saffron samples in terms of color, flavor, and bitterness. For instance, Greek saffron has been reported to contain a higher concentration of active ingredients compared to Indian saffron due to different drying processes and storage, which could lead to increased concentrations of glycoside carotenoids [37].

Additionally, as previously mentioned, environmental factors (altitude, temperature, and soil) may affect the crocin content of saffron [31,32], as well as the development of secondary metabolites during the plant growth [38]. The resulting phytochemical content could strongly influence the biological activity of saffron extracts. *C. sativus* L. grows well in Mediterranean and continental climates. The plant is capable of resisting temperatures ranging from −15 to 40 °C [31]. Altitude is a key factor in determining the content of crocin in saffron stigmas [39]: the higher the sun exposure and altitude, the higher the crocin content. In Greece, saffron is produced in the region of Kozani [40], which has an altitude of 600 m. In contrast, the Sicilian saffron comes from the city of Maletto, in the Etna area, located at an altitude of 1000 m. In Iran, saffron typically comes from the Qaen region, with an altitude between 1400 and 1500 m [41]. Crocin content is also strongly influenced by the composition of the soil: saffron prefers a variety of well-drained calcareous soils. To prevent water stagnation, cultivation is recommended on sloping soils to ensure good water runoff. In Iran, saffron is cultivated on sandy-textured calcareous soils, whereas in Greece, it grows on sandy clay soil with a pH of 7.4, and a high calcium carbonate content [42]. The amount of organic matter is extremely poor both in the Iranian and Greek areas. On the contrary, in the Etna area, the soil is volcanic, fertile, and rich in nutrients, organic matter and minerals (nitrates, phosphates, magnesium, potassium, and calcium).

All above-mentioned environmental characteristics could explain the different crocin content observed in the varieties of saffron under investigation.

As the coloring of saffron is related to the presence of carotenoids, especially crocin, the ISO 3632-1:2011 standard (last reviewed and confirmed in 2017 and currently in force), which establishes specifications for dried saffron obtained from the pistils of *C. sativus* L. flowers, was applied to obtain information on crocin content in the extracts from different variety of saffron. As this is a qualitative method based on coloring, it was coupled to UPLC-MS/MS analysis to quantify the amount of crocin in the sample. The obtained results showed that the Sicilian saffron (Figure 2) had the highest crocin content among the tested varieties. These data support the correlation between the carotenoid content and the cultivation area as carotenoid concentration rises with increasing sun exposure and altitude [43,44].

### 2.2. Antioxidant and NO Scavenging Activity of Saffron Extracts

The ORAC assay has been widely accepted as a suitable in vitro test to measure the antioxidant activity of natural active compounds [34] while the NO scavenger assay provides useful information about the efficiency in counteracting an overproduction of NO radicals, which has been associated with various pathological conditions, involving oxidative stress and DNA mutations.

As illustrated in Table 2, all tested samples showed excellent antioxidant activity compared to the reference standard Trolox. The ability of the different saffron extracts to inhibit the spontaneous production of NO from a sodium nitroprusside solution, determined via NO scavenger assay, decreased in the following order: Greek > Sicilian > Iranian.

According to the data reported in Table 1, Sicilian and Iranian saffron extracts were expected to provide the highest antioxidant activity due to their higher crocin content. On the contrary, the Sicilian variety was the least effective in the ORAC assay, thus suggesting that other extract components were involved in determining the radical scavenging efficiency. According to the literature [45], the main active ingredients of saffron are carotenoids (crocetins, crocins, α-carotene, β-carotene, lycopene, zeaxanthin, mangicrocin, xanthone-carotenoid), monoterpene aldehydes (picrocrocin, safranal and its isomers), monoterpenoids (crocusatins), and flavonoids (kaempferol derivatives). As spices containing phenolic and flavonoid compounds show antioxidant activities [46,47,48], the radical scavenging properties of saffron could be ascribed to both its phenolic content and active ingredients, such as safranal, crocin, crocetin, and carotene, whose antioxidant efficiency has been widely investigated [48]. A study by Makhlouf et al. [49] supported the importance of the phenolic content of saffron extracts in determining the protective effect against radical species for the different organs such as liver and heart.

Therefore, to explain the obtained results, the total phenolic content of each extract was determined using the Folin–Ciocalteu reactive. This method is used extensively but specific experimental conditions (e.g., sequence and timing of reagent additions, initial and final concentrations of sodium carbonate, incubation time and temperature, concentration of alcohols) could affect the results of the test, making the comparison of values obtained from different laboratories not entirely reliable [50]. However, when the same analysts in the same experimental conditions carry out the test, the obtained results allow reliable comparisons among different samples. As reported in Table 2, the extract of Greek saffron showed a value of total phenolic content (0.55 ± 0.12 mg GAE/100 g) about two-fold greater than that of the other two extracts, Sicilian and Iranian, whose values were almost similar (0.29 ± 0.10 and 0.23 ± 0.08 mg GAE/100 g, respectively). As natural extracts are made up of hundreds of components, a full characterization is hard to achieve. However, we attempted to identify the main constituents, aside from crocin. In Table 3, the molecules identified by UPLC-MS/MS analyses in each investigated sample are listed. The UPLC-MS/MS spectra of the three varieties of saffron extracts are shown in Figure 3.

As a quantitative determination of each ingredient contained in saffron extracts was beyond of the scope of this work, to compare the relative abundance of each molecule in the different extracts, the peak area was calculated for each ingredient from the UPLC–MS/MS spectrum (Table 3). It is interesting to note that no active ingredient showed a greater relative abundance in Greek saffron compared to that of both Iranian and Sicilian saffron. For instance, in Greek saffron, caffeic acid was more abundant than in Sicilian saffron but less abundant than in Iranian saffron. The opposite trend was observed comparing the relative abundance of α-carotene and β-carotene in the investigated saffron extracts: in Greek saffron, α-carotene and β-carotene were more abundant than in Iranian saffron but less abundant than in Sicilian saffron.

Different contents of active ingredients in saffron extracts of different origins have already been reported by others. Sobolev et al. [51] developed an NMR analytical protocol to evaluate the metabolite profiles of saffron extracts of different geographical origin (Greece, Spain, Hungary, Turkey, and Italy). The obtained results highlighted the influence of both environmental factors and crop production technology in determining the content of saffron constituents. The quantification via HPLC of major constituents of eleven different saffron sources showed significant differences among the tested samples but no correlation between saffron origin and active ingredient content could be hypothesized [37]. A study performed on saffron from seven different geographical area showed that crocin was the main active ingredient in all saffron extracts but the content of monosaccharides with antioxidant activity was significantly different in saffron samples from different sources [52].

The results of our work suggest that the antioxidant activity of the investigated extracts could not be related to a specific component, but it could be ascribed to the complex mixture of phytochemicals that naturally occur in each extract. The greater activity of saffron in comparison to crocin has already been reported by Asdaq and Inamdar [53], who studied their antihyperlipidemic and antioxidant potential after oral administration in rats. The authors concluded that other components than crocin were involved in determining the synergistic antihyperlipidemic and antioxidant potential of saffron, suggesting that the flavonoid content could be responsible for the better efficacy of saffron compared to crocin.

### 2.3. Antiglycation Activity

The in vivo accumulation of advanced glycation end products (AGEs), resulting from non-enzymatic reactions between proteins and reducing sugars (glycation), may lead to different cell and tissue damages [54,55]. As recent reports pointed out the effectiveness of several polyphenols in counteracting AGEs formation [56], we thought it would be interesting to evaluate the anti-glycation activity of the saffron extracts under investigation. As shown in Figure 4, Sicilian and Greek saffron extracts displayed similar activities against AGEs formation, providing about 40% inhibition, while the Iranian saffron was significantly (*p* < 0.05) less effective (30% inhibition). As saffron extracts contained several different polyphenols (see Table 3), the mechanisms underlying the observed anti-glycation activity were quite difficult to elucidate. As reported in the literature [56], some polyphenols, such as quercetin, can prevent AGEs production by inhibiting methylglyoxal formation while others, such as phenolic acids, can act as ROS inhibitors. Due to the presence of different types of polyphenols (e.g., quercetin, caffeic acid, p-coumaric acid, etc.) in the investigated saffron extracts, concurrent mechanisms could be hypothesized in determining the observed anti-glycation activity.

## 3. Materials and Methods

### 3.1. Chemicals

All solvents (ethanol, methanol LC/MS grade, water LC/MS grade, acetonitrile LC/MS grade, formic acid) and chemical compounds crocetin digentiobiose ester (crocin); fluorescein (FL); (2,2′-Azobis(2-methylpropionamidine) dihydrochloride (AAPH); 6-Hydroxy-2,5,7,8-tetramethylchroman-2-carboxylic acid (Trolox); aminoguanidine carbonate (AMG); bovine serum albumin (BSA); D-fructose; sodium nitroprusside; Folin and Ciocalteu’s phenol reagent; and Griess reagent (1% dihydrochloride sulphanylamide) were of analytical grade and were purchased from Merck (Milan, Italy).

### 3.2. Plant Material and Preparation of the Extracts

Sicilian saffron was purchased from the Capizzi Agricultural Company that provided a certificate claiming the Etna area as the origin of the saffron samples. Greek saffron from the Kozani region and Iranian saffron from the Qaen region were provided by a local herbalist shop that certified their origin. Samples were stored at −5 to 5 °C in the dark. Each extract was obtained from 10 g of stigmas in 200 mL of methanol at room temperature (r.t.) for 48 h. After filtration through Whatman^®^ Grade 1 filter paper (Whatman, UK), the obtained extracts were evaporated at 25 °C using a rotatory evaporator (Stuart RE300) under reduced pressure to obtain 0.98310 g (±0.1 mg) of dry extracts.

### 3.3. Determination of Crocin Content

Ultra-performance liquid chromatography coupled with mass spectrometry (UPLC-Ms/Ms) (Perkin-Elmer/AB SCIEX API 2000TM) was performed in order to analyze the content of crocin in saffron extracts. The separation was performed using water: acetonitrile with 0.1% formic acid as mobile phase with a gradient elution as follows: 0–3 min, 95:5 (*v*/*v*); 3–6 min, 70:30 (*v*/*v*); 6–10 min, 50:50 (*v*/*v*); 10–13, min 5:95 (*v*/*v*); and 13–30 min, 95:5 (*v*/*v*). The elution rate was 300 µL/min for 30 min into a C18 column (Phenomenex Kinetex^®^ 2.6 µm C18 100 Å, 100 × 2.1 mm, with a volume of injection of 10 µL. To comprehensively assess the analyzed spectrum, ESI-Ms/Ms was used with positive and negative polarities for detailed spectrum analysis, with the equipment settings as previously reported [57]. To quantify the crocin content in each variety of saffron, crocin was properly diluted in water and the calibration curve was constructed in the range 8000 ng/mL–150,000 ng/mL using the software’s “Quantitation Wizard” function. The limit of detection (LOD) of the analytical method was 0.812 ng/mL, while the limit of quantitation (LOQ) was 2.143 ng/mL.

As the coloring of saffron is related to the presence of carotenoids, especially crocin, saffron samples were additionally analyzed according to the ISO 3632-1:2011 standard [58]. Firstly, the homogenization of the stigmas (250 mg) with 500 mL of deionized water was carried out for 1 h. A total of 10 mL of the obtained mixture was diluted ten-fold with deionized water and filtered through a 0.45 µm filter. The absorbance was measured at 440 nm using a Perkin-Elmer Lambda 25 UV-Vis (Perkin Elmer, Waltham, MA, USA). This method was coupled to UPLC-MS/MS analysis to quantify the crocin content.

### 3.4. Determination of Total Phenolic Content

The polyphenol content of three saffron varieties was determined by the Folin–Ciocalteu method according to Aiyegoro and Okoh [59], with slight modifications. Firstly, samples (125 mg of dry stigma extract for each saffron variety) were prepared by adding a hydroalcoholic solution (50:50 *v*/*v*). A total of 5 mL of each sample was mixed with 2.5. mL of Folin–Ciocalteu reagent (diluted ten-fold with deionized water) and 2.5 mL of sodium bicarbonate (10% *w*/*v*). After incubation at 45 °C for 15 min, the absorbance of the samples was measured at 765 nm using a UV–vis spectrophotometer (Thermo Scientific Genesys 10 Scanning UV-Visible Spectrophotometer, Santa Clara, CA, USA). The standard calibration curve was prepared by dissolving gallic acid in water at the following concentrations: 0, 0.05, 0.1, 0.15, 0.2, and 0.25 mg·mL^−1^. The total phenolic content (TPC) was expressed as a milligram of gallic acid equivalents (GAE)/g of extract.

### 3.5. ORAC Assay

The antioxidant activity of saffron extracts was determined via the oxygen radical absorbance capacity (ORAC) assay, as previously described [60,61]. In particular, this assay measures the loss of fluorescence over time due to peroxyl-radical formation caused by the breakdown of AAPH (2,2′-azobis-2-methyl-propanimidamide, dihydrochloride). Trolox, a water-soluble vitamin E analogue, was used as a positive control, inhibiting fluorescein decay in a dose-dependent manner. Experiments were carried out using AAPH as peroxyl radical generator, while Trolox (12.5 µM) and phosphate buffer (pH 7.0) were used as standard and blank, respectively. A total of 50 µL of each extract was properly diluted (1 mg/mL), Trolox or buffer were placed in a 96-multiwell-plate, and the fluorescein solution (12 nM) was added. After preincubation for 30 min at 37 °C, AAPH solution (100 mM) was added to each well. Fluorescence was monitored using a VICTOR Wallac 1420 Multilabel Counters fluorimeter (Perkin Elmer, Boston, MA, USA), where excitation and emission wavelengths were 540 and 575 nm, respectively. Experiments were performed in triplicate. ORAC values were calculated using Origin^®^ 7 software (Origin Lab Corporation). ORAC units were calculated according to Equation (1) and expressed as µmol Trolox/µg sample:ORAC value (µmol/µg) = K(S sample − S blank)/(S Trolox − S blank)(1)
where K is the sample dilution factor and S is the area under the fluorescence decay curve of sample, Trolox, or blank. Each experiment was performed in triplicate and data were expressed as mean ± SD.

### 3.6. NO Scavenger Assay

The NO radical scavenging activity of saffron extracts was determined using a well-established procedure [62]. This colorimetric assay measures the antioxidant activity of a compound by evaluating its ability to inhibit the spontaneous production of NO from a sodium nitroprusside solution.

The reaction mixture, containing the sample under investigation (1mg/mL of saffron dried extract), phosphate buffer, and an aqueous solution of sodium nitroprusside (20 mM) was incubated at 25 °C for 150 min. After incubation, Griess reagent was added and the sample absorbance was measured at 540 nm with a spectrophotometer (Multiskan^®^ EX, Thermo Scientific, Waltham, MA, USA). The percentage of the inhibition of NO radical production was obtained using Equation (2):% of inhibition of NO = [A_0_ − A_1_] /A_0_ × 100(2)
where A_0_ is the absorbance of blank, while A_1_ is the absorbance of the sample.

### 3.7. Antiglycation Activity

The antiglycation activity of the saffron stigmas was determined by measuring their ability to inhibit the formation of fluorescent AGEs using the Maillard reaction [62,63]. Bovine serum albumin (BSA) (10 mg/mL) was incubated with D-fructose (0.5 M) in phosphate buffer 50 mM, pH 7.4, and NaN_3_ 0.02% *w*/*v* as positive controls. BSA alone was used as a negative control as it did not provide any formation of fluorescent AGEs. Aminoguanidine (AMG; 3 mM) was used as the reference compound. Final glycated BSA solution (300 µL) alone and with three different extracts (1 mg/ mL) was incubated at 37 °C in 96-well microtiter closed with their silicon lids for 7 days. Inhibition rate was determined (λ_exc_ 370 nm; λ_em_ 440 nm) using a VICTOR Wallac 1420 Multilabel Counter fluorimeter (Perkin Elmer, Waltham, MA, USA). The results were expressed as relative fluorescence units (RFU) and calculated according to Equation (3):% of inhibition = [1 − (RFU sample/RFU − positive control)] × 100(3)

### 3.8. Statistical Analysis

All data were expressed as mean ± standard deviation (±SD) of three replicates of three independent experiments. The statistical significance of these data was assessed using the one-way ANOVA test followed by Fisher’s method using the software package SYSTAT, version 11 (Systat Inc., Evanston, IL, USA). A *p*-value < 0.05 was considered statistically significant.

## 4. Conclusions

Although crocin content was higher in Sicilian saffron extracts, the highest in vitro antioxidant activity was observed for Greek saffron extracts, which showed the highest total phenolic content. All extracts demonstrated an interesting in vitro anti-glycation activity, which was higher for Greek and Sicilian saffron compared to Iranian saffron. The analysis of the relative abundance of saffron extract constituents suggested that the observed antioxidant and anti-glycation activity could be due to the complex mixture of phytochemicals in each extract rather than to the content of crocin or other specific components.

Therefore, saffron extracts deserve further investigations as supplements in food and pharmaceutical products to prevent or treat pathological processes induced by AGEs and radical species, considering that their effectiveness can be strongly affected by the saffron variety.

## Figures and Tables

**Figure 1 plants-12-03606-f001:**
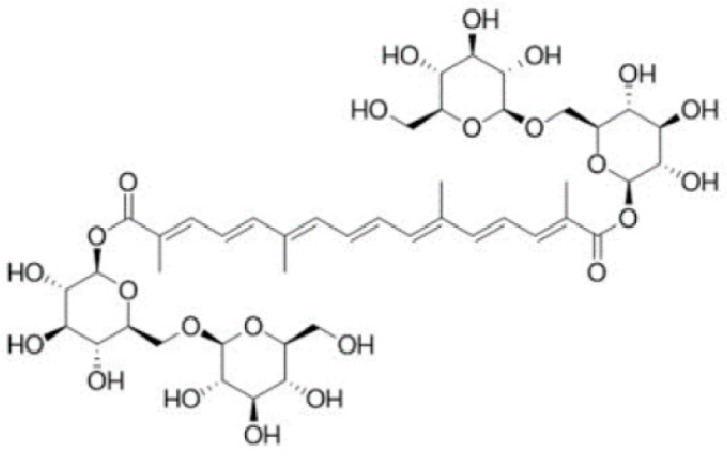
Chemical structure of crocin (all-*trans* crocetin di-β-D-gentiobiosyl ester).

**Figure 2 plants-12-03606-f002:**
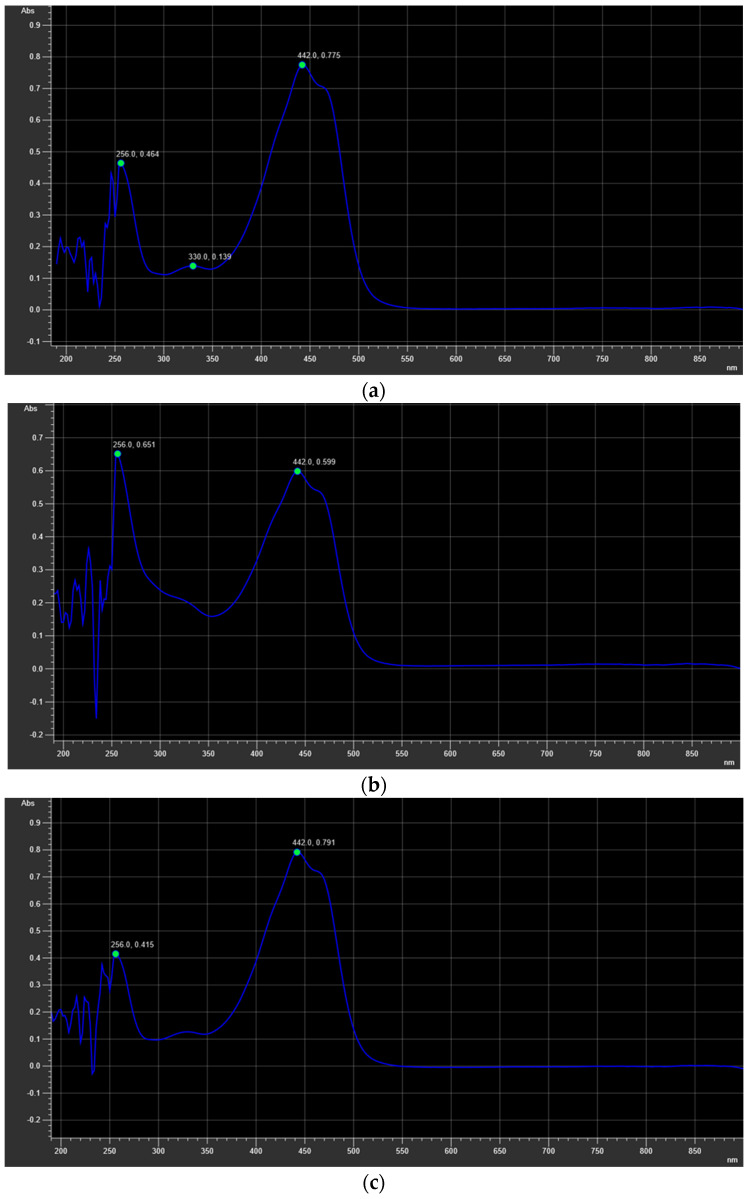
UV spectra of saffron extracts: (**a**) Sicilian extract; (**b**) Greek extract; (**c**) Iranian extract.

**Figure 3 plants-12-03606-f003:**
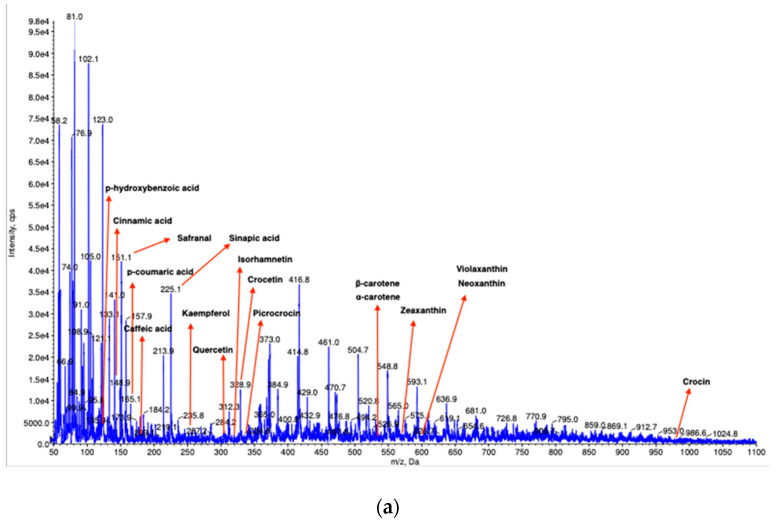
Tandem mass spectrum of saffron extracts in positive ion mode (*m*/*z* 100–1100 Da): (**a**) Sicilian saffron; (**b**) Greek saffron; (**c**) Iranian saffron.

**Figure 4 plants-12-03606-f004:**
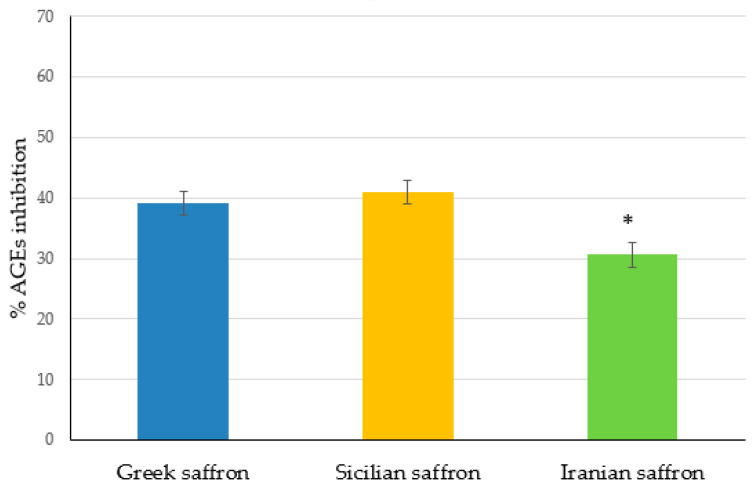
Percentage of AGEs inhibition of different saffron extracts (* *p* < 0.05 vs. Greek and Sicilian saffron).

**Table 1 plants-12-03606-t001:** Crocin concentrations in Greek, Sicilian, and Iranian saffron extracts. Data are expressed as mean of three experiments ± SD (* *p* < 0.05 vs. Sicilian and Iranian variety).

Saffron Variety	Crocin Content ng/mL
Greek	111,000 ± 2000 *
Sicilian	128,000 ± 6000
Iranian	126,000 ± 4000

**Table 2 plants-12-03606-t002:** Percentage of inhibition and antioxidant activity of Greek, Sicilian, and Iranian saffron samples. ORAC Units: Trolox equivalents for µM of sample; Trolox = 1 Units ORAC; data represent the mean of three independent experiments ± SD (* *p* < 0.05 vs. Sicilian and Iranian saffron; ** *p* < 0.05 vs. Greek and Iranian saffron; *** *p* < 0.05 vs. Greek saffron).

Variety	ORAC Units	% NO Inhibition	mg GAE/100 g
*Greek saffron*	50.9 ± 0.5 *	35.2 ± 0.2 *	0.55 ± 0.12 *
*Sicilian saffron*	14.7 ± 0.5 **	30.2 ± 0.2 **	0.29 ± 0.10 ***
*Iranian saffron*	26.5 ± 0.5	23.4 ± 0.2	0.23 ± 0.08 ***

**Table 3 plants-12-03606-t003:** Mass (*m/z*), retention time (RT), peak area, and relative abundance of the active ingredients identified in saffron extracts. SiS = Sicilian saffron; IrS = Iranian saffron; GrS = Greek saffron.

Compound	*m*/*z* (g/mol)	RT (min)	Peak Area			Relative Abundance (%)		
			SiS	IrS	GrS	SiS	IrS	GrS
Caffeic acid	180.16	0.51	4.72 × 10^4^	9.32 × 10^4^	7.37 × 10^4^	7%	15%	11%
Cinnamic acid	148.16	0.51	3.37 × 10^5^	5.05 × 10^5^	2.56 × 10^5^	50%	82%	39%
Crocetin	328.40	2.64	1.46 × 10^5^	2.04 × 10^5^	1.37 × 10^5^	22%	33%	21%
Isorhamnetin	316.26	2.64	2.70 × 10^4^	8.01 × 10^4^	2.62 × 10^4^	4%	13%	4%
Kaempferol	286.23	2.64	3.77 × 10^5^	1.11 × 10^5^	8.80 × 10^4^	56%	18%	13%
Neoxanthin	600.88	8.12	7.96 × 10^4^	4.47 × 10^4^	3.33 × 10^4^	12%	7%	5%
p-coumaric acid	164.04	0.51	2.29 × 10^4^	4.52 × 10^5^	3.86 × 10^4^	3%	74%	6%
p-hydroxybenzoic acid	138.12	n.r. *	3.19 × 10^4^	4.01 × 10^4^	2.42 × 10^4^	5%	7%	4%
Picrocrocin	330.37	2.64	3.19 × 10^4^	4.01 × 10^4^	2.42 × 10^4^	5%	7%	4%
Quercetin	302.23	n.r. *	1.01 × 10^4^	3.89 × 10^4^	9.64 × 10^3^	1%	6%	1%
Safranal	150.21	2.64	1.03 × 10^5^	9.32 × 10^4^	6.58 × 10^4^	15%	15%	10%
Sinapic Acid	224.21	2.64	4.11 × 10^5^	5.72 × 10^5^	2.85 × 10^5^	61%	93%	44%
Violaxanthin	600.85	8.12	1.91 × 10^4^	2.92 × 10^4^	1.78 × 10^4^	3%	5%	3%
Zeaxanthin	569.88	7.24	1.85 × 10^4^	2.38 × 10^4^	1.75 × 10^4^	3%	4%	3%
α-carotene	536.87	7.24	2.36 × 10^4^	1.91 × 10^4^	2.28 × 10^4^	3%	3%	3%
β-carotene	536.87	7.24	2.35 × 10^4^	1.90 × 10^4^	2.29 × 10^4^	3%	3%	4%

* n.r.: not reported due to low concentration in the extract.

## Data Availability

All data generated or analyzed during this study are included in this published article.

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
