# Peer review of "Evaluation of Crocin Content and In Vitro Antioxidant and Anti-Glycation Activity of Different Saffron Extracts"

_plants, 2023, doi:10.3390/plants12203606_

Round 1

Reviewer 1 Report

The manuscript entitled In vitro evaluation of antioxidant and anti-glycation activity of crocin from different saffron extracts” investigates the crocin content and the in vitro antioxidant activity of saffron extracts obtained from three different varieties (Greek, Sicilian, and Iranian saffron). The authors reported quantifying the crocin content in saffron extracts by ultra-performance liquid chromatography coupled with mass spectrometry. The antioxidant activity of saffron extracts was evaluated using the oxygen radical absorbance capacity (ORAC) assay and nitric oxide (NO) radical scavenging test. The Maillard reaction was used to assess the anti-glycation activity. Although the Sicilian and Iranian saffron extracts contained higher amounts of crocin than the Greek ones, ORAC values, and % NO inhibition were higher for the Greek variety, which showed a two-fold greater phenolic content than the other two extracts. Sicilian and Greek saffron had similar anti-glycation activities, while the Iranian saffron was less effective. These results suggest that the antioxidant activity of saffron extracts could be ascribed to their naturally occurring complex mixture of phytochemicals, deserving further investigations as a supplement to prevent pathological conditions induced by radical species.

The below revisions are recommended:

  1. The introduction is too lengthy. Concise it.

  1. The UV spectrum of the Sicilian saffron extract has been incorporated in the manuscript (Fig. 2). Please insert UV spectra of the Greek and Iranian saffron extracts.   

  1. What internal standard was used in UPLC? How was it selected?

  1. Table 3: Insert the raw UPLC/MS-MS chromatograms of the three different (Greek, Sicilian, and Iranian) varieties of saffron extracts in the text, indicating the mass, retention time, and abundance of each compound as described in Table 3. These three chromatograms are the key pieces of evidence for the authors’ claim.

  1. There are many reports available in the literature that include similar research. Cite some recent reports in the manuscript.

  1. The manuscript must be thoroughly checked, and the quality of the language must be improved. There are numerous grammatical mistakes.

  1. Uniformity (font and size) should be mentioned throughout the manuscript, including the schemes and figures. The authors are encouraged to check the journal IFA.

Please see above.

Author Response

Reviewer 1

The manuscript entitled “In vitro evaluation of antioxidant and anti-glycation activity of crocin from different saffron extracts” investigates the crocin content and the in vitro antioxidant activity of saffron extracts obtained from three different varieties (Greek, Sicilian, and Iranian saffron). The authors reported quantifying the crocin content in saffron extracts by ultra-performance liquid chromatography coupled with mass spectrometry. The antioxidant activity of saffron extracts was evaluated using the oxygen radical absorbance capacity (ORAC) assay and nitric oxide (NO) radical scavenging test. The Maillard reaction was used to assess the anti-glycation activity. Although the Sicilian and Iranian saffron extracts contained higher amounts of crocin than the Greek ones, ORAC values, and % NO inhibition were higher for the Greek variety, which showed a two-fold greater phenolic content than the other two extracts. Sicilian and Greek saffron had similar anti-glycation activities, while the Iranian saffron was less effective. These results suggest that the antioxidant activity of saffron extracts could be ascribed to their naturally occurring complex mixture of phytochemicals, deserving further investigations as a supplement to prevent pathological conditions induced by radical species.

The below revisions are recommended:

  1. The introduction is too lengthy. Concise it.

Answer

We would like to thank the reviewer for the valuable comments. As requested, we shortened the introduction as follows:

Crocin (all-trans crocetin di-β-D-gentiobiosyl ester) is a glycoside carotenoid (Figure 1) endowed with several health benefits including antioxidant, anticancer, antidepressant, anxiolytic, cardioprotective and neuroprotective activity [1]. Crocin belongs to the “crocin” family consisting of hydrophilic carotenoids in which D-glucose and/or D-gentiobiose residues form either mono- or di-glycosyl polyene esters of crocetin. It is noteworthy that the high glycosyl content makes crocins unusual water-soluble carotenoids [2]. 

The main sources of these natural compounds are the dried stigmas of Crocus sativus L. (commonly named saffron) and the fruits of Gardenia jasminoides E. Crocus sativus L. is a perennial herb, belonging to the Iridaceae family, and widely cultivated in the Mediterranean area and Western Asia, being Iran its main producer [3,4]. Due to its properties, saffron is one of the world’s most expensive and popular spices, widely used to add deep flavor and color to several kinds of dishes [5].

Nowadays, saffron is being widely investigated for its potential therapeutic applications. Many studies highlighted the ability of saffron extracts to act as antioxidant, anti-inflammatory, anticancer, wound healing, anti-ageing agents [6-13]. Clinical trials highlighted the efficiency of saffron capsules in ameliorating depression symptoms in comparison to placebo [14] or conventional drugs such as imipramine [15] and fluoxetine [16]. Similarly, clinical studies performed on patients with mild to moderate Alzheimer’s disease showed significantly better outcome on cognitive function after administering saffron capsules in comparison to placebo [17] while the same effectiveness was observed when saffron capsules were compared to donezepril [18]. Studies carried out on different animal models pointed out the efficiency of saffron and its active ingredients crocin and safranal in the treatment of pathological conditions such as atherosclerosis [19,20], cancer [21-23], hyperglycemia, and glucose uptake/metabolism [24].  It is interesting to note that Shirali et al. [25] reported the ability of crocin to lower significantly the formation of advanced glycation end-products, in addition to the levels of serum glucose, triglycerides, and total cholesterol in the diabetic rats. Goyal et al. [26] suggested the feasibility of using crocin as cardioprotective agent due to the improvement of cardiac functions in Wistar albino rats after crocin pretreatment. At cutaneous level, the involvement of crocin [27] and saffron extracts [28-30] in anti-inflammatory, antioxidant, and aging related processes has been reported.  

The quality of saffron and its content of phytochemicals, among which crocin plays a key role in determining the antioxidant activity, can be strongly affected by cultivation, territorial and climatic conditions [31,32]. Hence, it is crucial to evaluate the concentration of active ingredients according to the different saffron variety.  

In this context, the aim of this work was to assess the crocin content and the in vitro antioxidant activity of saffron extracts obtained from three varieties (Greek, Sicilian, and Iranian).  In vitro antioxidant activity of crocin has been already reported by others using different methods, such as DPPH (2,2′-diphenylpicryl hydrazyl free radical) radical scavenging assay [33], ethylene assay, and squalene peroxidation assay [30]. As DPPH is hydrophobic, the test requires the use of organic solvents [34], which have an adverse environmental impact. As crocin and saffron extracts are water-soluble, alternative methods, which do not require the use of organic solvents, could be carried out.  The ethylene assay and the squalene peroxidation assay assess hydroxyl radical and singlet oxygen scavenging activity, respectively [30]. Both these methods were developed to point out the antioxidant ability of topical formulations [35,36] and they were not validated on plant extracts. Therefore, we assessed the antioxidant activity of saffron extracts using the oxygen radical absorbance capacity (ORAC) assay, which is regarded as a reliable tool for screening antioxidants [34]. Additionally, in vitro nitric oxide (NO) radical scavenging ability of the saffron extracts under investigation was evaluated. As oxidative stress may result in the formation of advanced glycation end products (AGEs), a further aim of this study was to determine the anti-glycation activity of saffron extracts using the Maillard reaction.

  1. The UV spectrum of the Sicilian saffron extract has been incorporated in the manuscript (Fig. 2). Please insert UV spectra of the Greek and Iranian saffron extracts. 

Answer

As requested, we inserted UV spectra of Greek and Iranian saffron extracts in Fig. 2. 

  1. What internal standard was used in UPLC? How was it selected?

Answer

We did not use an internal standard. Calibration was carried out utilizing an external standard, as described in Section 3.3. To quantify the crocin content in each saffron variety, the crocin standard was appropriately diluted in water and a calibration curve was constructed in the range of 8,000–150,000 ng/mL. As the aim of our work was to determine the crocin content in different saffron extracts, the crocin standard purchased from Merck was utilized in this study to evaluate crocin content.

  1. Table 3: Insert the raw UPLC/MS-MS chromatograms of the three different (Greek, Sicilian, and Iranian) varieties of saffron extracts in the text, indicating the mass, retention time, and abundance of each compound as described in Table 3. These three chromatograms are the key pieces of evidence for the authors’ claim.

Answer

As requested by the reviewer, we included in the text the UPLC-MS/MS spectra of the three varieties of saffron extracts. We inserted in Table 3 the mass, relative abundance, and retention time of each compound.

  1. There are many reports available in the literature that include similar research. Cite some recent reports in the manuscript.

Answer

As requested by the reviewer, we cited some works reporting data about different varieties of saffron.

We inserted in the manuscript the following citations:

“A different content of active ingredients in saffron extracts of different origins has already been reported by others. Sobolev et al. [51] developed an NMR analytical protocol to evaluate the metabolite profiles of saffron extracts of different geographical origin (Greece, Spain, Hungary, Turkey, and Italy). The obtained results highlighted the influence of both environmental factors and crop production technology in determining the content of saffron constituents. Quantification by HPLC of major constituents of eleven different saffron sources showed significant differences among the tested samples but no correlation between saffron origin and active ingredient content could be hypothesized [52]. A study performed on saffron from seven different geographical area showed that crocin was the main active ingredient in all saffron extracts but the content of monosaccharides with antioxidant activity was significantly different in saffron samples from different sources [53].” 

The cited references were inserted in the reference list as follows:

  1. Cicco, N.; Lanorte, M.T.; Paraggio, M.; Viggiano, M.; Lattanzio, V. A reproducible, rapid and inexpensive Folin–Ciocalteu micro-method in determining phenolics of plant methanol extracts. Microchem J. 2009, 91, 107-110.
  2. Sobolev, A.P.; Carradori, S.; Capitani, D.; Vista, S.; Trella, A.; Marini, F.; Mannina L. Saffron Samples of Different Origin: An NMR Study of Microwave-Assisted Extracts. Foods. 2014, 3, 403-419.
  3. Caballero-Ortega, H.; Pereda-Miranda, R.; Abdullaev, F.I. HPLC quantification of major active components from 11 different saffron (Crocus sativus L.) sources. Food Chem. 2007, 100, 1126-1131.
  4. Zhang, A.; Shen, Y.; Cen, M.; Hong, X.; Shao, Q.; Chen, Y.; Zheng, B. Polysaccharide and crocin contents, and antioxidant activity of saffron from different origins. Ind Crops Prod. 2019, 133, 111-117.
  5. The manuscript must be thoroughly checked, and the quality of the language must be improved. There are numerous grammatical mistakes.

Answer

As requested, the manuscript was checked for the quality of the language. A native English speaker revised the manuscript.

  1. Uniformity (font and size) should be mentioned throughout the manuscript, including the schemes and figures. The authors are encouraged to check the journal IFA.

Answer

We used the template provided by the Journal Plants to prepare the manuscript. As requested by the reviewer, we checked that font and size were in accordance with those required by the Journal Plants.

Reviewer 2 Report

The work by Ronsisvalle and colleagues addressed the antioxidant and anti-glycation activities of croci from different saffron extracts. In general the manuscript is well-written and has potential to be published in the Plants MDPI journal. There are some aspects that should be properly addressed before this manuscript is considered for publication:

- abstract: some numerical data should be added in the results section

- introduction should be revised and reorganized in a more concise manner; it is a little ambiguous and some of the data presented here should be placed in the discussion section while comparing the data obtained in this work with that already published by other authors

- results and discussion section: from lines 110-133 the data presented is not directly related to the major objective of this work; it is important to emphasize that geographical conditions markedly affect the chemical composition and consequently the biological activity of plant extracts but such findings were not properly highlighted in the results section. Chemical composition was presented but the quantification and relative abundance of each of the chemical constituents identified was not presented. Equally important to underline is that proper references to authors that have already chemically characterized such extracts should be added in order to clarify if the relative abundances and peak areas identified are overlapping. Finally, and contrarily to what authors have stated in the lines 186-187, quantitative determination is crucial, and thus should be presented.

- materials and methods: being the different matrices from commercial origin, how can the authors ensure that they were properly identified? to what concerns to the assessment of the antioxidant activity, what was the criteria for selecting such in vitro tests? statistical analysis section should be fully revised and completed, and the software used as well as the different tests applied should be described here.

- conclusion section should be fully rewritten; it is a mere repetition of the results obtained; it is expected to have a brief summary sentence and also future perspectives, as well as the translational focus of the findings stated here.

There are some typos and incongruences in phrase construction.

Author Response

Reviewer 2

The work by Ronsisvalle and colleagues addressed the antioxidant and anti-glycation activities of croci from different saffron extracts. In general the manuscript is well-written and has potential to be published in the Plants MDPI journal. There are some aspects that should be properly addressed before this manuscript is considered for publication:

  1. Abstract: some numerical data should be added in the results section

Answer

We would like to than the reviewer for the valuable comments. As requested, we added some numerical data in the results section of the abstract. At line 21, we added the following data: “Although the Sicilian and Iranian saffron extracts contained higher amounts of crocin (128 ± 6 ng/mL and 126 ± 4 ng/mL, respectively) compared to the Greek one (111 ± 2 ng/mL), ORAC values (50.9 ± 0.5) and % NO inhibition (35.2 ± 0.2) were higher for the Greek variety that showed a total phenolic content about two-fold greater than that of the other two extracts.

  1. Introduction should be revised and reorganized in a more concise manner; it is a little ambiguous and some of the data presented here should be placed in the discussion section while comparing the data obtained in this work with that already published by other authors.

Answer

As requested, the Introduction was shortened and some sentences were modified as follows:

Crocin (all-trans crocetin di-β-D-gentiobiosyl ester) is a glycoside carotenoid (Figure 1) endowed with several health benefits including antioxidant, anticancer, antidepressant, anxiolytic, cardioprotective and neuroprotective activity [1]. Crocin belongs to the “crocin” family consisting of hydrophilic carotenoids in which D-glucose and/or D-gentiobiose residues form either mono- or di-glycosyl polyene esters of crocetin. It is noteworthy that the high glycosyl content makes crocins unusual water-soluble carotenoids [2]. 

The main sources of these natural compounds are the dried stigmas of Crocus sativus L. (commonly named saffron) and the fruits of Gardenia jasminoides E. Crocus sativus L. is a perennial herb, belonging to the Iridaceae family, and widely cultivated in the Mediterranean area and Western Asia, being Iran its main producer [3,4]. Due to its properties, saffron is one of the world’s most expensive and popular spices, widely used to add deep flavor and color to several kinds of dishes [5].

Nowadays, saffron is being widely investigated for its potential therapeutic applications. Many studies highlighted the ability of saffron extracts to act as antioxidant, anti-inflammatory, anticancer, wound healing, anti-ageing agents [6-13]. Clinical trials highlighted the efficiency of saffron capsules in ameliorating depression symptoms in comparison to placebo [14] or conventional drugs such as imipramine [15] and fluoxetine [16]. Similarly, clinical studies performed on patients with mild to moderate Alzheimer’s disease showed significantly better outcome on cognitive function after administering saffron capsules in comparison to placebo [17] while the same effectiveness was observed when saffron capsules were compared to donezepril [18]. Studies carried out on different animal models pointed out the efficiency of saffron and its active ingredients crocin and safranal in the treatment of pathological conditions such as atherosclerosis [19,20], cancer [21-23], hyperglycemia, and glucose uptake/metabolism [24].  It is interesting to note that Shirali et al. [25] reported the ability of crocin to lower significantly the formation of advanced glycation end-products, in addition to the levels of serum glucose, triglycerides, and total cholesterol in the diabetic rats. Goyal et al. [26] suggested the feasibility of using crocin as cardioprotective agent due to the improvement of cardiac functions in Wistar albino rats after crocin pretreatment. At cutaneous level, the involvement of crocin [27] and saffron extracts [28-30] in anti-inflammatory, antioxidant, and aging related processes has been reported.  

The quality of saffron and its content of phytochemicals, among which crocin plays a key role in determining the antioxidant activity, can be strongly affected by cultivation, territorial and climatic conditions [31,32]. Hence, it is crucial to evaluate the concentration of active ingredients according to the different saffron variety.  

In this context, the aim of this work was to assess the crocin content and the in vitro antioxidant activity of saffron extracts obtained from three varieties (Greek, Sicilian, and Iranian).  In vitro antioxidant activity of crocin has been already reported by others using different methods, such as DPPH (2,2′-diphenylpicryl hydrazyl free radical) radical scavenging assay [33], ethylene assay, and squalene peroxidation assay [30]. As DPPH is hydrophobic, the test requires the use of organic solvents [34], which have an adverse environmental impact. As crocin and saffron extracts are water-soluble, alternative methods, which do not require the use of organic solvents, could be carried out.  The ethylene assay and the squalene peroxidation assay assess hydroxyl radical and singlet oxygen scavenging activity, respectively [30]. Both these methods were developed to point out the antioxidant ability of topical formulations [35,36] and they were not validated on plant extracts. Therefore, we assessed the antioxidant activity of saffron extracts using the oxygen radical absorbance capacity (ORAC) assay, which is regarded as a reliable tool for screening antioxidants [34]. Additionally, in vitro nitric oxide (NO) radical scavenging ability of the saffron extracts under investigation was evaluated. As oxidative stress may result in the formation of advanced glycation end products (AGEs), a further aim of this study was to determine the anti-glycation activity of saffron extracts using the Maillard reaction.

  1. Results and discussion section: from lines 110-133 the data presented is not directly related to the major objective of this work; it is important to emphasize that geographical conditions markedly affect the chemical composition and consequently the biological activity of plant extracts but such findings were not properly highlighted in the results section. Chemical composition was presented but the quantification and relative abundance of each of the chemical constituents identified was not presented. Equally important to underline is that proper references to authors that have already chemically characterized such extracts should be added in order to clarify if the relative abundances and peak areas identified are overlapping. Finally, and contrarily to what authors have stated in the lines 186-187, quantitative determination is crucial, and thus should be presented.

Answer

To comply with the reviewer’s request, we added the following sentence: “The resulting phytochemical content could strongly affect the biological activity of saffron extracts.”

In the revised version of the manuscript, the relative abundance of each component was determined and reported in Table 3. As reported in the literature (Alavizadeh SH, Hosseinzadeh H. Bioactivity assessment and toxicity of crocin: a comprehensive review. Food Chem Toxicol. 2014;64:65-80. doi:10.1016/j.fct.2013.11.016), the extraction method can strongly affect the phytochemical content of saffron extracts. Therefore, comparisons among extracts obtained using different extraction methods could hardly provide overlapping results. Hence, we believe that comparing the results of works performed using different experimental conditions could not provide the reader with useful information as it would be a hard task to understand if differences are due to the extraction procedures or to the origin of saffron.

As stated in the manuscript, a quantitative determination of each ingredient contained in saffron extracts was out of scope of our work. We agree with the reviewer that a quantitative determination is crucial but it is generally performed when quality control processes are involved. In addition, such quantitative determination is carried out only on specific extract constituents to standardize the extract. In this work, we report an in vitro study whose results show a good antioxidant and anti-glycation activity of the Greek saffron extract that could be worthy of further in vivo investigations. At present, as we do not have data supporting the biological activity of the investigated extract, we believe it would be untimely to standardize the extracts.

  1. Materials and methods: being the different matrices from commercial origin, how can the authors ensure that they were properly identified? to what concerns to the assessment of the antioxidant activity, what was the criteria for selecting such in vitro tests? statistical analysis section should be fully revised and completed, and the software used as well as the different tests applied should be described here.

Answer

As reported in section 3.2, the supplier certified the origin of saffron. The authors were provided with proper certifications about saffron origin.

As crocin and saffron extracts are water soluble, we chose the ORAC assay to assess the antioxidant activity because this assay did not require the use of organic solvents. 

As requested, the statistical analysis section was revised and the software used as well as the different tested applied were inserted. The statistical analysis section was revised as follows: “All data were expressed as mean ± standard deviation (±SD) of three replicates of three independent experiments. The statistical significance of these data was assessed by the one-way ANOVA test followed by Fisher's method using the software package SYSTAT, version 11 (Systat Inc., Evanston IL, USA). A p-value < 0.05 was considered statistically significant.”

  1. Conclusion section should be fully rewritten; it is a mere repetition of the results obtained; it is expected to have a brief summary sentence and also future perspectives, as well as the translational focus of the findings stated here.

Answer

In the conclusion section, we summarized the main results to provide the reader with an overview of the work reported in the manuscript. We would like to discuss the translational focus of the findings of our work but we did not perform any test to evaluate the capacity of the different saffron varieties used in our study to conclude they hold specific bioactivities. Therefore, we prefer not to draw conclusions on potential health effects of saffron from the results we obtained because such conclusions would be mere speculations. The fact that the investigated extracts had in vitro radical scavenging activity (as shown by ORAC data) does not guarantee that, when administered in humans, these effects are going to be preserved or that such extracts will improve defense mechanisms.

In the revised version of the manuscript, we shortened the repetition of the results. Conclusion section was rewritten as follows:

Although crocin content was higher in Sicilian saffron extracts, the highest in vitro antioxidant activity was observed for Greek saffron extracts, which showed the highest total phenolic content.  All extracts demonstrated an interesting in vitro anti-glycation activity, which was higher for Greek and Sicilian saffron compared to Iranian saffron. The analysis of the relative abundance of saffron extract constituents suggested that the observed antioxidant and anti-glycation activity could be due to the complex mixture of phytochemicals in each extract rather than to the content of crocin or other specific components.

Therefore, saffron extracts would deserve further investigations as supplement in food and pharmaceutical products to prevent or treat pathological processes induced by AGEs and radical species, considering that their effectiveness could be strongly affected by the saffron variety.

  1. There are some typos and incongruences in phrase construction.

Answer

We apologize for the typos. The manuscript was checked for the quality of the language. A native English speaker revised the manuscript.

Reviewer 3 Report

The article studied the in vitro antioxidant activity and the antiglycation activity of saffron extracts obtained from three different varieties. However, the data fails to explain why the Sicilian saffron has the highest crocin content but least effective in the ORAC assay. It also fails to explain why the three varieties have different antiglycation activity. The paper does not have enough data to explain the observed scientific questions, so it should be rejected. 

Line 70,what does “old” mean?

Line 86-87, why the DPPH method is not suitable for crocin? Is crocin soluble in methanol? I think the DPPH method works ok in methanol.

Figure 2,can you add the data for other two varieties?

Line 189, “none of the identified active ingredients in the extract of Greek saffron was more abundant than in the Sicilian and Iranian saffron extracts”. This mention is wrong. Greek saffron has more caffeic acid than Sicilian saffron, and more carotene than Iranian saffron. In some cases, the active ingredients contents of Greek saffron are very similar as Sicilian saffron.

Author Response

Reviewer 3

The article studied the in vitro antioxidant activity and the antiglycation activity of saffron extracts obtained from three different varieties. However, the data fails to explain why the Sicilian saffron has the highest crocin content but least effective in the ORAC assay. It also fails to explain why the three varieties have different antiglycation activity. The paper does not have enough data to explain the observed scientific questions, so it should be rejected. 

Answer

We would like to thank the reviewer for taking the time to review our manuscript.

Reviewer 3 states, “the data fails to explain why the Sicilian saffron has the highest crocin content but least effective in the ORAC assay. It also fails to explain why the three varieties have different antiglycation activity”.

In the results and discussion section, we provided the following explanations:

  1. a) ….. these results suggest that the antioxidant activity of the investigated extracts could not be related to a specific component, but it could be ascribed to the complex mixture of phytochemicals that naturally occur in each extract. A greater activity of saffron in comparison to crocin has already been reported by Asdaq and Inamdar [54] studying their antihyperlipidemic and antioxidant potential after oral administration in rats. The authors concluded that other components, apart from crocin, were involved in determining the synergistic antihyperlipidemic and antioxidant potential of saffron, suggesting that the flavonoid content could be responsible for the better efficacy of saffron compared to crocin.
  2. b) As saffron extracts contained several different polyphenols (see Table 3), the mechanisms underlying the observed anti-glycation activity were quite difficult to elucidate. As reported in the literature [57], some polyphenols, such as quercetin, can prevent AGEs production by inhibiting methylglyoxal formation while others, such as phenolic acids, can act as ROS inhibitors. Due to the presence of different types of polyphenols (e.g. quercetin, caffeic acid, p-coumaric acid etc.) in the investigated saffron extracts, concurrent mechanisms could be hypothesized in determining the observed anti-glycation activity.
  3. Line 70,what does “old” mean?

Answer

The word “old” was a typo that was deleted in the revised version.

  1. Line 86-87, why the DPPH method is not suitable for crocin? Is crocin soluble in methanol? I think the DPPH method works ok in methanol.

Answer

We did not state that the DPPH method was not suitable for crocin. We stated, “As DPPH is hydrophobic and requires the use of organic solvents to perform the test [34], such method could not be regarded as the most suitable to evaluate the antioxidant activity of crocin and/or saffron extracts.” As DPPH is hydrophobic, the test requires the use of organic solvents, which have an adverse environmental impact. As crocin and saffron extracts are water-soluble, alternative methods, which do not require the use of organic solvents, could be carried out. Therefore, the DPPH method could not be regarded as the most suitable test to determine the antioxidant activity of crocin and saffron extracts. To better elucidate the meaning of the previous statement, we modified the text as follows: “As DPPH is hydrophobic, the test requires the use of organic solvents [34], which have an adverse environmental impact. As crocin and saffron extracts are water-soluble, alternative methods, which do not require the use of organic solvents, could be carried out.”

  1. Figure 2,can you add the data for other two varieties?

Answer

We added the data for Greek and Iranian saffron extracts in Fig. 2.

  1. Line 189, “none of the identified active ingredients in the extract of Greek saffron was more abundant than in the Sicilian and Iranian saffron extracts”. This mention is wrong. Greek saffron has more caffeic acid than Sicilian saffron, and more carotene than Iranian saffron. In some cases, the active ingredients contents of Greek saffron are very similar as Sicilian saffron.

Answer

Due to the reviewer’s comment, we realized that the meaning of the sentence “none of the identified active ingredients in the extract of Greek saffron was more abundant than in the Sicilian and Iranian saffron extracts” could be ambiguous. Therefore, for a better comprehension, we rephrased the previous sentence as follows:

It is interesting to note that no active ingredient showed a greater relative abundance in Greek saffron compared to that of both Iranian and Sicilian saffron. For instance, in Greek saffron, caffeic acid was more abundant than in Sicilian saffron but less abundant than in Iranian saffron. The opposite trend was observed comparing the relative abundance of α-carotene and β-carotene in the investigated saffron extracts: in Greek saffron, α-carotene and β-carotene were more abundant than in Iranian saffron but less abundant than in Sicilian saffron.

Round 2

Reviewer 1 Report

The authors have revised the manuscript following my suggestions. I recommend the manuscript for publication.

Author Response

We would like to thank the reviewer for reviewing our manuscript. 

Reviewer 2 Report

Accept

Minor aspects should be revised

Author Response

(The authors gave the same response as above.)

Reviewer 3 Report

As you mentioned “these results suggest that the antioxidant activity of the investigated extracts could not be related to a specific component, but it could be ascribed to the complex mixture of phytochemicals that naturally occur in each extract”, “As saffron extracts contained several different polyphenols (see Table 3), the mechanisms underlying the observed anti-glycation activity were quite difficult to elucidate”. You did not have any specific conclusions about your goal. You cannot cite other's data to make conclusions in your study. You should provide more evidence and new findings to publish this article.

Also for the DPPH method, it is just an analytical method. It is recommended to use green method to extraction, but we need to use the most accurate method to analyze our sample. I am not sure if DPPH is better than ORAC, but your reason for choosing ORAC is not convinced. 

Author Response

Reviewer 3

As you mentioned “these results suggest that the antioxidant activity of the investigated extracts could not be related to a specific component, but it could be ascribed to the complex mixture of phytochemicals that naturally occur in each extract”, “As saffron extracts contained several different polyphenols (see Table 3), the mechanisms underlying the observed anti-glycation activity were quite difficult to elucidate”. You did not have any specific conclusions about your goal. You cannot cite other's data to make conclusions in your study. You should provide more evidence and new findings to publish this article.

Answer

In this work, we determined crocin content in three different varieties of saffron (Iranian, Greek and Sicilian). We reported the results of these analyses in Table 1. We assessed the antioxidant activity of each extract and we reported the obtained data in Table 2. As the extracts that contained the highest percentage of crocin (Sicilian and Iranian saffron) did not show the highest antioxidant activity, we determined the total phenolic content of each extract (Table 2) and the relative abundance of each identified ingredient (Table 3). We observed that the extract with the highest total phenolic content (Greek saffron) provided the highest antioxidant activity. However, no active ingredient showed a greater relative abundance in Greek saffron compared to that of both Iranian and Sicilian saffron. Therefore, we concluded that “the antioxidant activity of the investigated extracts could not be related to a specific component, but it could be ascribed to the complex mixture of phytochemicals that naturally occur in each extract”.

We drew this conclusion according to the data we reported in our manuscript.  We did not make conclusions citing other’s data.

Also for the DPPH method, it is just an analytical method. It is recommended to use green method to extraction, but we need to use the most accurate method to analyze our sample. I am not sure if DPPH is better than ORAC, but your reason for choosing ORAC is not convinced.

Answer

Organic solvents may have an environmental impact regardless of their use as extraction solvent, reactive, synthesis medium and so on. Therefore, if a valid alternative method is available, we prefer to avoid the use of organic solvents. We are sorry if the reviewer does not share our point of view.
